# High Temperature Sulfate Minerals Forming on the Burning Coal Dumps from Upper Silesia, Poland

**Jan Parafiniuk * and Rafał Siuda**

Faculty of Geology, University of Warsaw, Żwirki i Wigury 93, 02-089 Warszawa, Poland; rsiuda@uw.edu.pl
* Correspondence: j.parafiniuk@uw.edu.pl

**Abstract:** The subject of this work is the assemblage of anhydrous sulfate minerals formed on burning coal-heaps. Three burning heaps located in the Upper Silesian coal basin in Czerwionka-Leszczyny, Radlin and Rydułtowy near Rybnik were selected for the research. The occurrence of godovikovite, millosevichite, steklite and an unnamed $MgSO_4$, sometimes accompanied by subordinate admixtures of mikasaite, sabieite, efremovite, langbeinite and aphthitalite has been recorded from these locations. Occasionally they form monomineral aggregates, but usually occur as mixtures practically impossible to separate. The minerals form microcrystalline masses with a characteristic vesicular structure resembling a solidified foam or pumice. The sulfates crystallize from hot fire gases, similar to high temperature volcanic exhalations. The gases transport volatile components from the center of the fire but their chemical compositions are not yet known. Their cooling in the near-surface part of the heap results in condensation from the vapors as viscous liquid mass, from which the investigated minerals then crystallize. Their crystallization temperatures can be estimated from direct measurements of the temperatures of sulfate accumulation in the burning dumps and studies of their thermal decomposition. Millosevichite and steklite crystallize in the temperature range of 510–650 °C, $MgSO_4$ forms at 510–600 °C and godovikovite in the slightly lower range of 280–450 (546) °C. These values are higher than those previously reported.

**Keywords:** burning coal-dumps; sulfate minerals; exhalations; godovikovite; steklite; millosevichite

## 1. Introduction

Sulfate minerals are commonly associated with a supergene environment as products of processes taking place on the Earth's surface under low temperature conditions. The vast majority of the more than 300 known sulfate minerals are formed in two environments: as weathering products of oxidized sulfides and other ore minerals, and due to evaporation of sea water or salt lakes. Also well-known are also such sulfates as barite, less often celestine and anhydrite or gypsum crystallizing from hydrothermal solutions of rather low to medium temperatures. Numerous sulfate minerals are also found in volcanic exhalation products, where they crystallize from volcanic vapors over a wide temperature interval from several hundred to below 100 °C. These lower temperature minerals, such as members of the alunite group, halotrichite, alunogen, etc., are also known from the weathering zone of ores.

The fires of exposed coal seams or coal mines and burning coal-dumps located next to collieries in coal basins in many places around the world are interesting environments for the formation of sulfate minerals [1]. Fires initiating spontaneously in the interior of heaps generate streams of hot gases from which several dozen sulfate phases crystallize on or near the dump surface. They are generally similar to those of volcanic exhalation products. Sulfate phases forming on burning coal dumps have been described from Russia, Czech Republic, Germany, Poland, USA and other countries ([1] and literature therein) but the conditions for their crystallization are still insufficiently understood. Mineralogical studies were not favored by the fact that until recently they were treated as anthropogenic phases.

Currently, following the Commission on New Minerals, Nomenclature and Classification of the International Mineralogical Association (CNMNC IMA) decision, the naturally forming phases crystallizing on the burning coal dumps have obtained the status of real minerals, which should favor an intensification of research [2]. The limitation, however, is the diminishing availability of these environments due to the liquidation of heaps and the closure of coal mines in many countries. Due to their solubility in water, the discussed sulfate minerals are available for study only on heaps with active fire centers inside them and they quickly disappear after the fire is extinguished.

The focus of this work are anhydrous sulfates that are the first to crystallize from high-temperature fire gases on heaps of several collieries in the Upper Silesian coal basin near Rybnik, Poland. The main aim is to try to determine the conditions of their formation based on their mode of occurrence, chemical composition and thermal behavior. The results of these analyses were supplemented with field observations and gas temperature measurements.

## 2. Materials and Methods

The mineral samples for this study were collected on selected burning coal-dumps in the Upper Silesian coal basin, South Poland. Nowadays, mineral-forming processes can be mostly observed on the dumps in the western part of the coal basin in the Rybnik region, therefore our research has been limited to this area. Three burning heaps located in Czerwionka-Leszczyny, Radlin and Rydułtowy were selected for mineralogical study (Figure 1). It is worth emphasizing that no industrial slags or municipal waste were deposited in the heaps studied. They contain only rock material extracted from underground mine workings. Observations of the formation of exhalation minerals in these heaps have been carried out for over decade and have covered the periods of the highest activity of fires in their interior. Currently, the fires of these heaps are in a declining phase. Archival samples, kept in collections for over a dozen years or freshly collected, were used for the analysis.

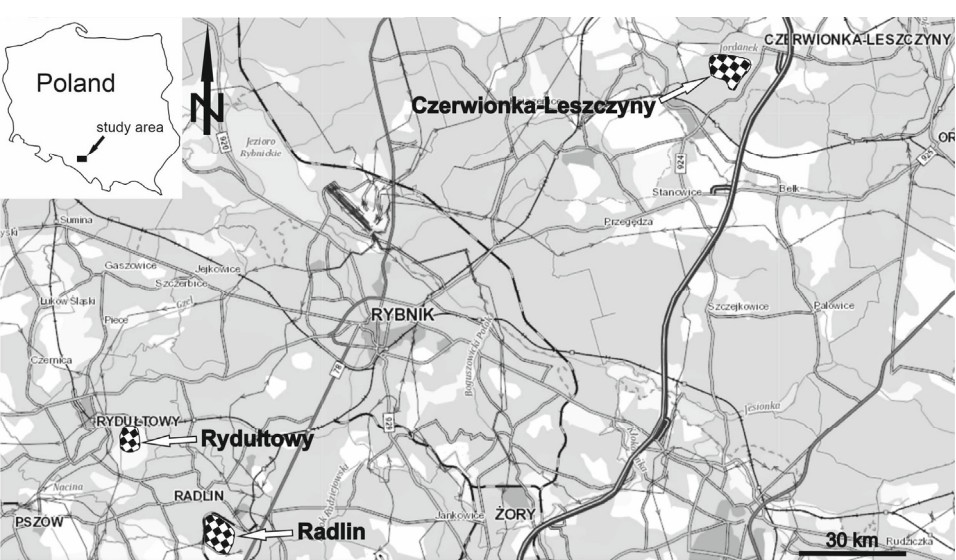

**Figure 1.** Location of the studied burning coal-dumps of Upper Silesia in the Rybnik region.

Powder X-ray diffraction (PXRD) and scanning electron microscope (SEM) with energy dispersive spectroscopy (EDS) attachment were used to identify minerals. The presence of all minerals was confirmed by PXRD using a powder X-ray diffractometer X'Pert PRO MPD (Almelo, The Netherlands). The parameters of the X-ray beam were as follows: CoKα wavelength, emitted from the X-ray tube with 40 mA and 40 kV current applied. X-ray patterns were recorded in the 2.5° to 75.99° 2θ range, with a step size of 0.02°. The results were processed using the X-ray analysis software X'Pert Plus HighScore (ver. 2.2e) and

ICDD PDF-2 database. SEM-EDS analysis were carried out using the electron microscope ZEISS SIGMA VP (Carl Zeiss Microscopy, Cambridge, UK). Microprobe chemical analyses and thermal analyses were also performed for selected samples. The chemical composition of sulfate minerals was determined by the wavelength-dispersive spectrometry electron probe microanalysis method (EPMA) using a CAMECA SX-100 electron microprobe (St. Denice, France). The analyses were done under a 15 keV accelerating voltage, 10 nA beam current and spot size ranging from 5 to 20 μm. Counting times of 20 s at peak and 10 s for background were applied. The pure analytical results were finally recalculated using ZAF correction procedures. Natural minerals and synthetic compounds supplied by Cameca and Structure Probe, Inc. were used during calibration of the microprobe. Analytical lines used and mean detection limits, expressed in wt %, were as follows: Na (albite, TAP, K$\alpha$, 0.04), Ca (diopside, TAP, K$\alpha$, 0.04), Mg (diopside, TAP, K$\alpha$, 0.02), Si (diopside, TAP, K$\alpha$, 0.03), Al (orthoclase, TAP, K$\alpha$, 0.03), K (orthoclase, PET, K$\alpha$, 0.02), Ti (rutile, LPET, K$\alpha$, 0.02), Fe (hematite, LIF, K$\alpha$, 0.13), S (barite, PET, K$\alpha$, 0.11), P (YPO$_4$, PET, K$\alpha$, 0.02). The thermogravimetry and differential scanning calorimetry (TGA/DSC) measurements were performed using a Mettler-Toledo TGA/DSC STAR$^e$ system (Schwerzenbach, Switzerland). Heating rates of 10 °C/min under a dry N$_2$ atmosphere and at a constant flow (50 mL/min) over a range of temperature from 25 to 1000 °C were applied respectively. Obtained data were analyzed using the STARe software provided by Mettler Toledo (Schwerzenbach, Switzerland). The total weight of each sample was accurately weighted into standard 70 μL alumina crucible using Mettler-Toledo XS105 DualRange balance (Schwerzenbach, Switzerland).

## 3. Anhydrous Sulfate Minerals from the Burning Coal-Dumps

### 3.1. Godovikovite NH$_4$Al(SO$_4$)$_2$

The most common anhydrous sulfate in the exhalation products on burning coal heaps in Upper Silesia is godovikovite. This mineral was first described by Shcherbakova et al. [3] from Kopeysk, South Ural Mountains, Russia and approved by the IMA although it was a product of a coal mine heap fire. Named in honor of Aleksandr Aleksandovich Godovikov (1927–1995), eminent Russian mineralogist and head of the Fersman Mineralogical Museum in Moscow, godovikovite is also known from burning coal dumps from Kladno [4] and Radvanice [5,6], both Czech Republic; from Alsdorf and Freital, Germany [7]; Pécs-Vasas, Hungary [8] and other places as well from burning coal seams in Wuda, Inner Mongolia, China [9] and Ravat, Tajikistan [10]. It has also been found in volcanic exhalation products in Solfatara di Pozzuoli, Campania and La Fossa crater, Vulcano Island, Italy but there it is rare and forms smaller aggregates compared to burning heaps.

Significant amounts of godovikovite occur on the burning heaps of the Upper Silesian Coal Basin, where they were found in Czerwionka-Leszczyny, Radlin and Rydułtowy [11]. As in other locations, godovikovite forms finely crystalline masses of white, sometimes grayish or yellowish, color. Godovikovite is the main component of the so-called sulfate crust formed just below the dump surface in places where fire gases escape. It is accompanied in variable proportions by other sulfate or chloride minerals: millosevichite, ammonioalunite–ammoniojarosite, sal ammoniac and others. The sulfate crust can reach locally considerable size, covering an area of several square meters and reaching a thickness of 10–20 cm (Figure 2). It takes the form of irregular layer of sulfates that stick together fragments and crumbs of shale rocks into a hard compact piece. Occasionally, you can see vents piercing through the crusts that mark escape routes of gases into the atmosphere. Sometimes inside the crust there are cavities formed by rock fragments, the surfaces of which are covered with solidified accumulations of godovikovite. Spectacular forms resembling cave stalactites hang from the ceiling of the larger cavities (Figure 3).

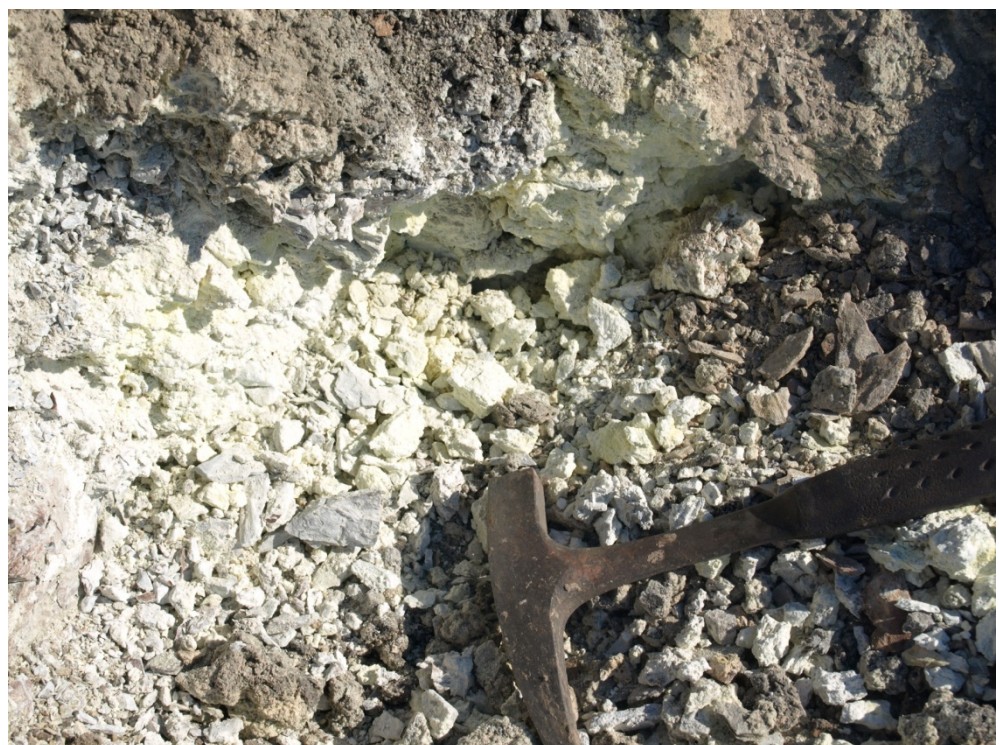

**Figure 2.** Freshly exposed sulfate crust composed mainly of godovikovite, Rydułtowy. Notice the yellowish color that fades quickly.

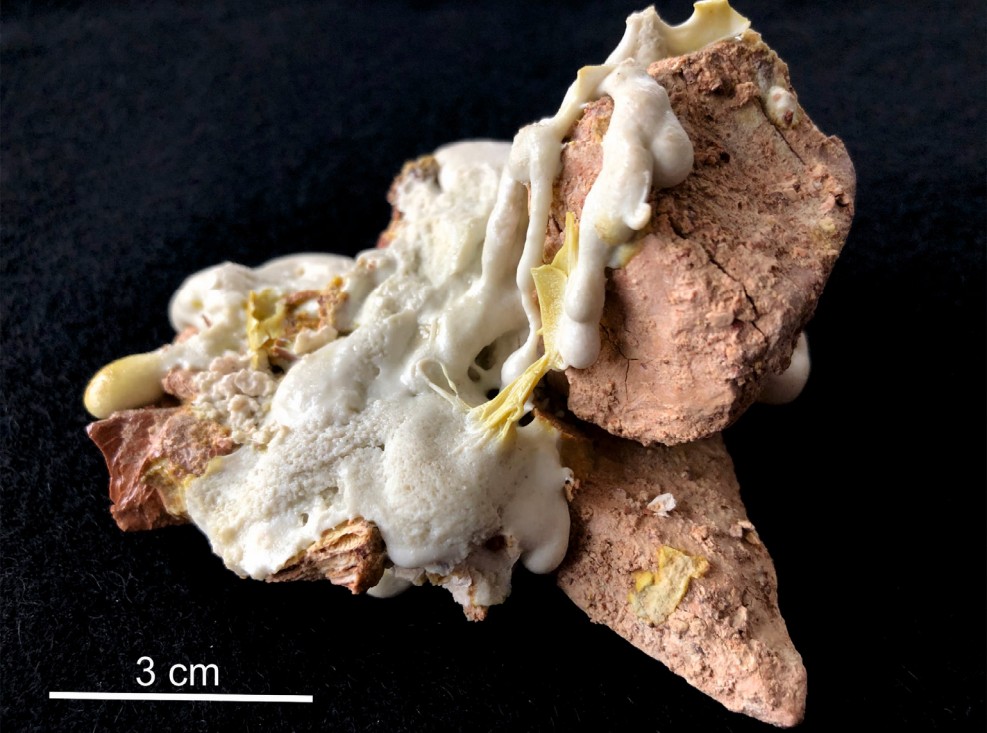

**Figure 3.** Dripstone aggregates of godovikovite generated from the semi-liquid masses.

In our opinion they were created from a hot, plastic, semi-liquid and gas-saturated mass of godovikovite which flowed down from the walls of the cavern. Such solidified masses of godovikovite have smooth surfaces, but their interior, like the entire sulfate crust, is porous, though to a lesser extent than, e.g., aggregates of millosevichite. The more

porous structure reveals smaller, more monomineralic godovikovite aggregates sometimes found on pieces of burned barren rock. The mineral is very finely crystalline and only in SEM image are visible its platy hexagonal crystals, which are at most a few μm in size. Godovikovite is rather unstable under ambient conditions. Samples kept in room humidity conditions are slowly hydrated and after few months transform into a more water-soluble tschermigite $NH_4Al(SO_4)_2 \cdot 12H_2O$ from the alum group. Tschermigite on heaps often accompanies or completely replaces godovikovite accumulations. Hence, godovikovite can be found only on heaps with an active fire center inside them. After the fire ceases, the mineral is quickly hydrated and then dissolved and carried away.

### 3.2. Millosevichite $Al_2(SO_4)_3$

Anhydrous aluminum sulfate-millosevichite has been discovered in volcanic exhalation products in Grotta dell'Alume and Grotte de Faraglione in Porto Levante on Vulcano Island, Italy and named in honor of Italian mineralogist Federico Millosevich (1875–1942) [12]. In the type locality the mineral forms granular masses of red or violet-blue color from a significant $Fe^{3+}$ admixture replacing Al; hence its chemical formula is often given as $(Al,Fe^{3+})_2(SO_4)_3$. More often than in volcanic exhalations millosevichite can be found on burning coal dumps. It was reported from Russia, Germany, Czech Republic, USA, Hungary and other places, similar to godovikovite occurrences. We have found this mineral many times in Radlin and Rydułtowy as a component of the sulfate crust or as separate aggregates in the hottest parts of the heaps but it is less common than godovikovite. Its aggregates, a few centimeters in size, sometimes larger, are white in color, which distinguishes them from the volcanic environment. Rarer are yellowish or pink varieties colored with fine hematite particles spread over the mass of millosevichite. Millosevichite from the burning heaps of Upper Silesia has a composition similar to theoretical aluminum sulfate and contains slight iron substitutions (Table 1).

A characteristic feature of the millosevichite aggregates from this environment, emphasized by all researchers, is their vesicular structure. All aggregates of millosevichite are very light, highly spongy and form thin films around the oval, small, hollow bubbles after fire gases (Figure 4). The mineral looks like solidified foam deposited on barren rock fragments or sticking together the slate fragments. Millosevichite is so finely crystalline that even under large magnification in SEM images individual crystals can hardly be observed. On the heaps, millosevichite can be found only in the zones of outflow of hot fire gases. After cooling down, the mineral remains on the heap for only a few months. The mineral is slowly dissolved in water and preserved under ambient conditions after a few months completely hydrates to alunogen $Al_2(SO_4)_3 \cdot 17H_2O$.

**Table 1.** Electron probe microanalyses (EPMA) of selected sulfates and their mixture from the Upper Silesian burning dumps (wt %).

| Sample Type | $Al_2O_3$ | $Fe_2O_3$ | MgO | CaO | $Na_2O$ | $K_2O$ | $TiO_2$ | $SO_3$ | $P_2O_5$ | $SiO_2$ | Total |
|---|---|---|---|---|---|---|---|---|---|---|---|
| | 22.59 | 4.69 | 1.55 | 0.36 | 0.25 | 1.94 | 0.46 | 56.26 | 0.15 | 3.74 | 92.00 * |
| Godovikovite | 24.95 | 4.63 | 1.61 | 0.46 | 0.25 | 0.99 | 0,55 | 53.33 | 0.23 | 3.93 | 90.92 * |
| with traces of | 20.87 | 5.98 | 1.27 | 1.33 | 0.22 | 1.53 | 0,55 | 57.20 | 0.19 | 0.72 | 89.84 * |
| millosevichite | 21.71 | 6.11 | 1.35 | 0.43 | 0.22 | 1.04 | 0.53 | 55.11 | 0.16 | 0.28 | 86.93 * |
| (sample 225A) | 23.72 | 4.46 | 1.02 | 0.41 | 0.19 | 0.94 | 0.58 | 54.61 | 0.20 | 0.36 | 86.50 * |
| | 23.27 | 5.74 | 1.47 | 0.40 | 0.20 | 0.95 | 0.53 | 58.68 | 0.18 | 0.54 | 91.96 * |
| | 29.01 | 1.09 | 4.21 | 0.49 | 1.40 | 0.29 | 0.21 | 60.69 | 0.34 | 0.10 | 97.83 |
| | 29.06 | 1.19 | 4.33 | 0.23 | 0.97 | 0.27 | 0.19 | 63.02 | 0.35 | 0.38 | 99.99 |
| | 30.75 | 1.14 | 5.00 | 0.24 | 1.52 | 0.34 | 0.18 | 57.73 | 0.39 | 0.01 | 97.29 |
| Millosevichite | 28.84 | 1.13 | 4.44 | 0.29 | 0.85 | 0.22 | 0.20 | 63.25 | 0.34 | 0.05 | 99.59 |
| with $MgSO_4$ | 28.23 | 1.90 | 4.29 | 0.26 | 1.33 | 0.28 | 0.20 | 61.59 | 0.34 | 0.24 | 98.67 |
| (sample R743) | 27.98 | 1.28 | 4.14 | 0.21 | 0.61 | 0.26 | 0.20 | 61.84 | 0.37 | 3.58 | 100.49 |
| | 30.12 | 1.22 | 4.46 | 0.22 | 0.69 | 0.19 | 0.19 | 60.60 | 0.35 | 0.15 | 98.19 |
| | 23.94 | 3.33 | 4.99 | 0.71 | 1.83 | 2.02 | 0.20 | 54.96 | 0.41 | 3.85 | 96.26 |
| | 23.14 | 7.48 | 3.65 | 0.56 | 1.09 | 0.73 | 0.33 | 58.07 | 0.21 | 0.15 | 95.42 |

**Table 1.** *Cont.*

| Sample Type | Al$_2$O$_3$ | Fe$_2$O$_3$ | MgO | CaO | Na$_2$O | K$_2$O | TiO$_2$ | SO$_3$ | P$_2$O$_5$ | SiO$_2$ | Total |
|---|---|---|---|---|---|---|---|---|---|---|---|
| | 14.22 | 4.14 | 1.95 | 2.69 | 1.31 | 11.29 | 0.00 | 48.49 | 0.31 | 0.55 | 84.96 |
| | 20.33 | 4.79 | 2.64 | 2.32 | 0.76 | 9.50 | 0.00 | 57.78 | 0.39 | 0.39 | 98.91 |
| Steklite | 19.71 | 3.70 | 1.92 | 4.17 | 0.89 | 9.07 | 0.00 | 53.05 | 0.32 | 0.71 | 93.54 |
| (sample R741) | 16.77 | 4.32 | 2.29 | 3.83 | 1.08 | 10.78 | 0.00 | 55.78 | 0.36 | 0.72 | 95.92 |
| | 18.94 | 4.83 | 3.03 | 1.83 | 1.35 | 10.47 | 0.00 | 54.59 | 0.42 | 0.79 | 96.24 |
| | 23.16 | 4.46 | 1.54 | 2.41 | 0.56 | 7.68 | 0.00 | 54.39 | 0.39 | 0.20 | 94.81 |
| | 14.32 | 4.28 | 9.55 | 0.90 | 1.06 | 9.71 | 0.00 | 58.11 | 0.45 | 0.92 | 99.31 |
| | 14.71 | 4.23 | 11.11 | 0.49 | 1.02 | 7.62 | 0.00 | 58.47 | 0.38 | 0.98 | 98.99 |
| | 13.80 | 4.13 | 10.41 | 0.75 | 1.02 | 8.94 | 0.00 | 58.03 | 0.32 | 0.64 | 98.04 |
| | 13.80 | 4.58 | 11.92 | 0.47 | 1.14 | 6.44 | 0.00 | 60.24 | 0.27 | 0.49 | 99.35 |
| Steklite with | 11.50 | 4.41 | 10.39 | 0.37 | 2.69 | 9.13 | 0.00 | 58.54 | 0.30 | 0.62 | 97.95 |
| MgSO$_4$ | 12.42 | 4.17 | 8.82 | 0.68 | 1.63 | 11.15 | 0.00 | 58.52 | 0.29 | 0.85 | 98.53 |
| (sample R700A) | 12.13 | 5.16 | 9.90 | 0.36 | 2.18 | 7.96 | 0.00 | 58.59 | 0.58 | 0.33 | 97.18 |
| | 4.84 | 10.58 | 8.95 | 1.24 | 1.27 | 9.67 | 0.00 | 56.55 | 0.17 | 0.30 | 93.57 |
| | 4.42 | 10.10 | 8.77 | 1.26 | 1.36 | 9.83 | 0.00 | 57.14 | 0.24 | 0.53 | 93.65 |
| | 4.66 | 9.87 | 8.77 | 1.34 | 1.44 | 10.38 | 0.00 | 53.96 | 0.14 | 0.07 | 90.63 |
| | 5.89 | 8.94 | 7.40 | 1.14 | 2.05 | 12.12 | 0.00 | 55.83 | 0.31 | 0.07 | 93.75 |

\* Without ammonium content.

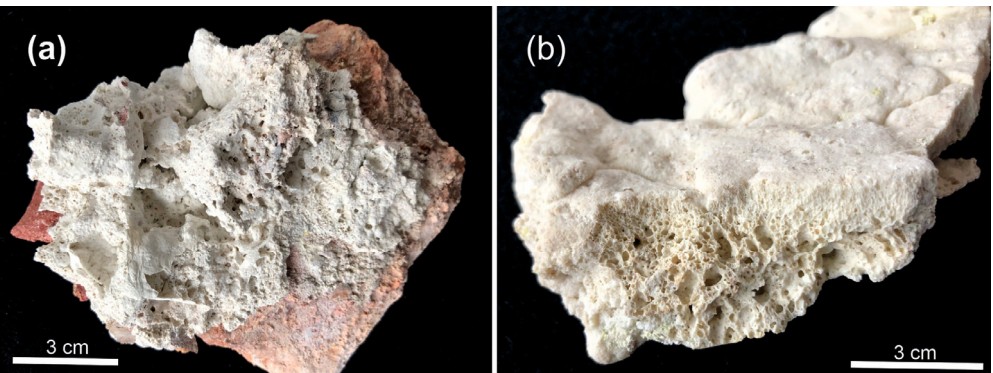

**Figure 4.** Millosevichite from Rydułtowy. (**a**) Monomineral accumulation covered red clinker fragment. (**b**) Vesicular structure typical for high temperature sulfates.

### 3.3. Steklite KAl(SO$_4$)$_2$

The anhydrous equivalent of alum-(K) known as steklite has been repeatedly recorded from burning heaps or coal seams affected by fire. Lapham et al. [13] described it without a name as KAl(SO$_4$)$_2$ from an anthracite seams fire in Pennsylvania. It was named steklite from Russian word steklo = glass by Chesnokov et al. [14] because its aggregates found at burning coal-dumps in Kopeysk near Chelyabinsk, Russia, resembled glass. This name was published without IMA approval. At that time the IMA did not recognize the phases resulting from fires as minerals and treated them as of anthropogenic origin. Only after it was found in exhalations of the Tolbachik volcano, Great Fissure eruption, Kamchatka, Russia, was it validated as a mineral species keeping the name steklite [15]. In volcanic exhalations of Kamchatka steklite forms microscopic transparent hexagonal plates with euchlorine. On burning heaps, steklite is also less common than godovikovite or millosevichite but, compared to volcanic exhalations, it forms much larger aggregates of up to 10 cm in size. In Upper Silesia, we found it in Radlin and Rydułtowy, in the hottest exhalation zones. It occurs there alone or admixed with millosevichite, sometimes also with other anhydrous sulfates from which it is macroscopically indistinguishable. Sulfate mineral mixtures can be so fine crystalline that they are difficult to identify by routine methods, including Raman spectroscopy recommended by Košek et al. [16]. Steklite is developed very similarly to millosevichite as white, strongly vesicular, finely crystalline masses re-

sembling solidified foam or pumice. The individual steklite crystals can sometimes be seen under high magnification in SEM images. They are developed as tiny to a few μm plates with hexagonal outline (Figure 5). The mineral is not deliquescent but preserved in room condition for a few weeks is hydrated by air moisture to alum-(K). On the dumps steklite is stable only in hot parts close to the dump surface over the active fire.

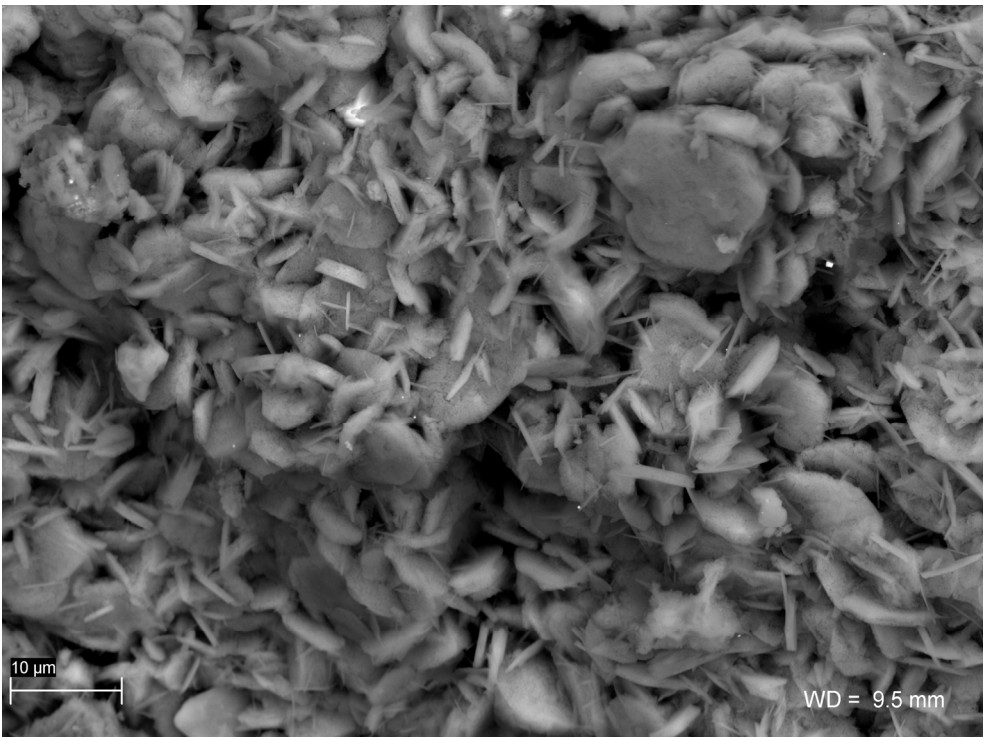

**Figure 5.** SEM image of platy microcrystals of steklite from Rydułtowy.

### 3.4. Unnamed Anhydrous Magnesium Sulfate MgSO₄

The least commonly reported mineral of high-temperature sulfate assemblages on burning coal dumps is anhydrous magnesium sulfate. So far, it has been mentioned only from Kopeysk near Chelyabinsk, South Ural Mountains, Russia [17] and is still very little known. A phase with this chemical composition is not on the list of officially recognized minerals. On the other hand, minerals representing hydrated magnesium sulfates from $MgSO_4 \cdot H_2O$ (kieserite) to $MgSO_4 \cdot 7H_2O$ (epsomite) are well known. Anhydrous $MgSO_4$ has been found by us on burning heaps in the Rybnik region in Radlin and Rydułtowy in the hottest parts of the exhalations. Sporadically it forms monomineral aggregates visually indistinguishable from millosevichite or steklite. They have the typical appearance of high-temperature sulfates from this environment and form foamed, vesicular masses of white or yellowish color (Figure 6). This phase more often occurs as a very fine admixture in millosevichite or steklite aggregates where it can only be detected using a microprobe. Anhydrous $MgSO_4$ is probably more common on burning heaps than we have found out, but it is very difficult to identify. The presence of this phase cannot be detected with the use of PXRD, which is routinely used in mineralogical practice as it is x-ray amorphous. According to van Essen et al. [18] amorphous $MgSO_4$ can recrystallize above 276 °C but only when a small amount of water is present (0.2 water molecule per $MgSO_4$ molecule). We did not observe such recrystallization in our samples. Anhydrous $MgSO_4$ is not stable in ambient conditions and quickly hydrates, first to kieserite $MgSO_4 \cdot H_2O$ and then to more hydrated phases. Kieserite, possibly also crystallizing as the primary phase from fire gases at lower temperatures or by evaporation of water solution, is known from many burning coal-heaps [8,17,19]. It is also quite common in Upper Silesia, recorded at Czerwionka-Leszczyny, Radlin and Rydułtowy. The more strongly hydrated magnesium sulfates are of

an obviously secondary nature. Hexahydrite $MgSO_4·6H_2O$ is the most commonly found mineral of this group on the burning coal heaps. Its accumulation in the form of white, powdery or fluffy blooms can cover an area of several square meters on the cooled parts of heaps. Epsomite $MgSO_4·7H_2O$ is rare and is only found in areas with high humidity. In dry, sunny places it quickly dehydrates to hexahydrite. Even more rare is starkeyite $MgSO_4·4H_2O$, sometimes found as a minor component of the sulfate assemblages.

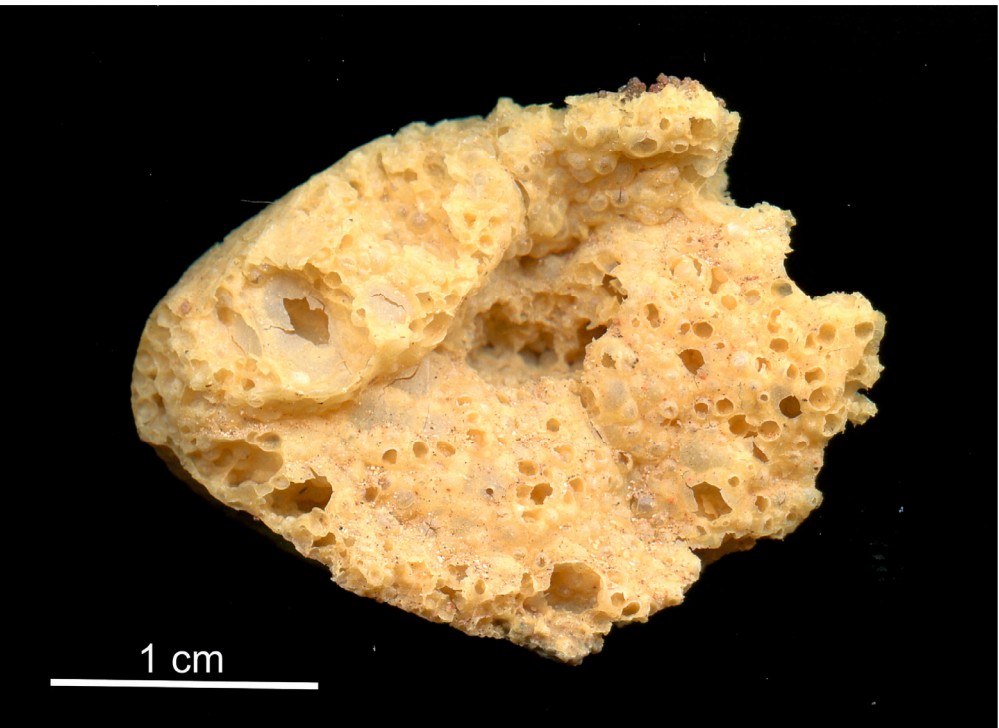

**Figure 6.** Vesicular mass of almost pure, XRD-amorphous $MgSO_4$ from Radlin.

*3.5. Other Anhydrous Sulfates*

The four minerals described above are sometimes accompanied by small amounts of other anhydrous sulfates. They are minor components of finely crystalline mixtures of sulfates from which they cannot be separated. So far, it has not been possible to find separate clusters in the Upper Silesia heaps, so no more detailed description can be given. Some of them, however, occur in greater amounts on other burning heaps, and are therefore typical of this environment. The iron analogue of godovikovite is sabieite $NH_4Fe(SO_4)_2$. Although it was first found in the Lone Creek Falls cave near Sabie, RSA, as a product of pyrite weathering [20], it is known from many burning heaps [4,8,21], and oil shale fires [22]. In Upper Silesia, it was sporadically recorded as a minor component of the sulfate crust in Czerwionka-Leszczyny, Radlin and Rydułtowy.

Efremovite $(NH_4)_2Mg_2(SO_4)_3$ completes the list of anhydrous ammonium sulfates known from the high temperature exhalations on burning coal-dumps. This mineral was described from a dump at Kopeysk in the Chelyabinsk area and named in honor of Russian geologist Ivan Antonovich Yefremov (1908–1972) [23]. It is also known from many other burning heaps including Czerwonka-Leszczyny, Radlin and Rydułtowy [11], although it does not occur anywhere in larger accumulations. One reason may be that efremovite quickly hydrates to boussingaultite $(NH_4)_2Mg(SO_4)_2·6H_2O$, more common in this environment.

Efremovite in our localities accompanies in small amounts godovikovite in the sulfate crusts. A mineral better known and more common than efremovite is its potassium counterpart langbeinite $K_2Mg_2(SO_4)_3$. Langbeinite is mainly thought to be a constituent of marine or salt lakes evaporates and is known from many K-Mg salt deposits around the world.

This mineral has also been noted from Kamchatka fumaroles and from burning coal-dumps in Chelyabinsk coal basin [17] and Kladno and Radvanice, Czech Republic [6]. We have found it as a minor addition in the steklite accumulations in Rydułtowy. Aphthitalite $(K,Na)_3Na(SO_4)_2$ is another anhydrous sulfate better known from marine or lacustrine salt deposits than from volcanic exhalations although its type locality is Mt. Vesuvius, Campania, Italy. As its name suggests (aphthitos = unalterable) the mineral is stable in ambient conditions. We found aphthitalite as a small admixture in the sulfate crust composed of millosevichite and godovikovite in Rydułtowy (Figure 7). There are two other anhydrous sulfates forming commonly in the burning heap environment: mascagnite $(NH_4)_2SO_4$ and anhydrite $CaSO_4$, but these are formed at lower temperatures and will not be considered here.

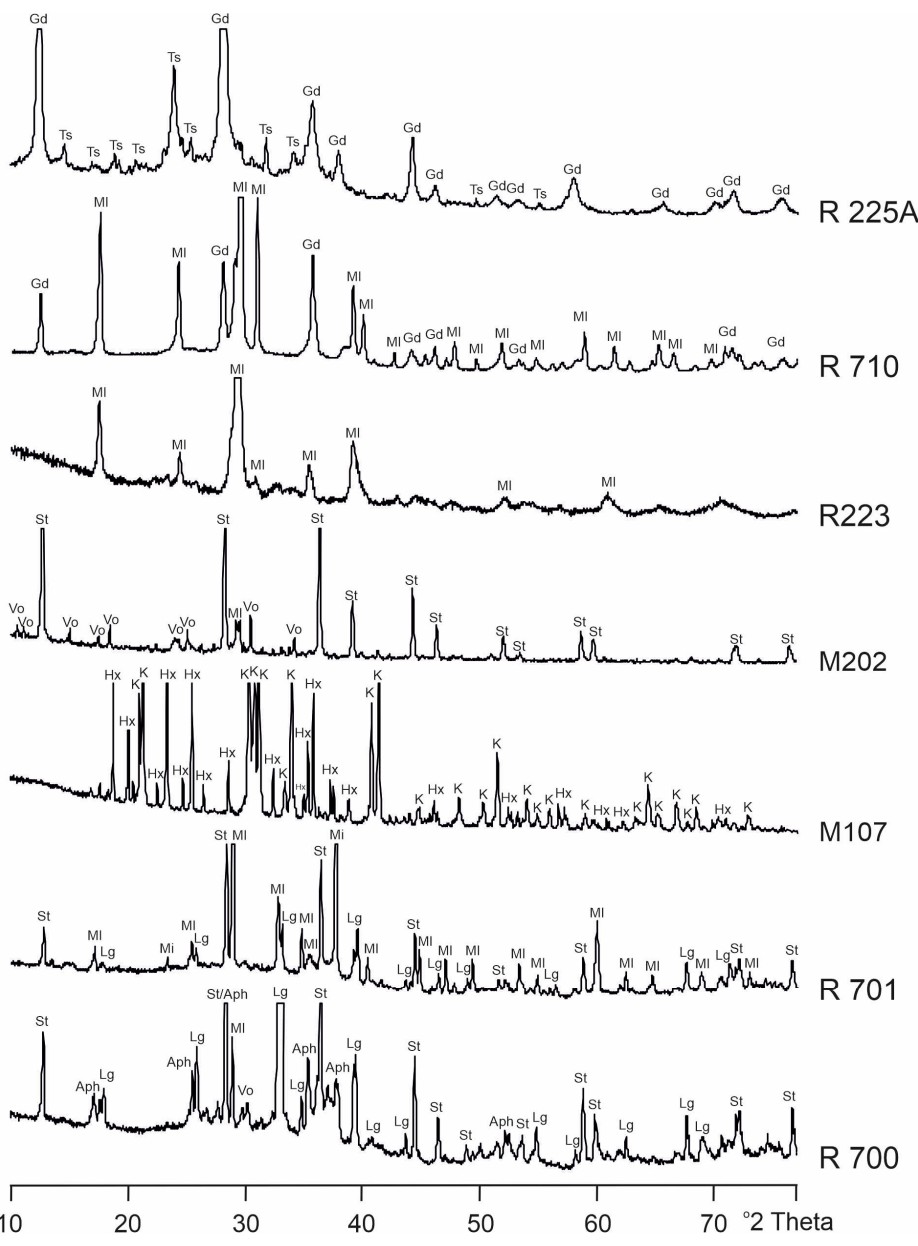

**Figure 7.** Selected XRD patterns of anhydrous sulfates from the Upper Silesian burning coal-dumps. Gd—godovikovite, Ts—tschermigite, Ml—millosevichite, St—steklite, Vo—voltaite, K—kieserite, Hx—hexahydrite, Aph—aphthitalite, Lg—langbeinite.

## 4. Conditions for the Crystallization of Assemblage of High Temperature Sulfates

Mineral-forming processes related to the exhalation of hot fire gases and occurring in the near-surface parts of heaps are very complex. They lead to the formation of overlapping mineral assemblages. Separating them is usually difficult, and a closer determination of the conditions of their genesis still awaits a solution. It is also not favored by the fact that mineral assemblages are usually made up of mixtures of fine crystalline minerals, some of which are only of limited stability, and most of which are soluble in water.

A relatively simple process to distinguish here is sublimation, or more precisely resublimation, i.e., crystallization of minerals directly from the gas phase. In this way, sulfur, salammoniac, sometimes selenium or some crystalline hydrocarbons like ravatite (phenanthrene) can be formed, although sulfur and selenium can also crystallize from the liquid phase. Sublimation minerals are usually formed as acicular or skeletal aggregates that grow around gas vents. In general, the crystallization temperatures of these minerals are not high. Therefore, they can be indicators of the initial stages of a fire inside the heap marked by the crystallization of salammoniac and sulfur, or fire extinguishing, which is associated with the crystallization of ravatite or other aromatic hydrocarbons. The crystallization temperature of salammoniac can be determined at 115–125 °C, monoclinic sulfur is formed above 96 °C, usually after salammoniac crystallization. Ravatite crystallizes at a temperature of about 40 °C [21].

The process that explains the formation of most exhalative minerals is vapor–liquid–solid (VLS) crystallization. VLS is widely used in experimental mineral syntheses and in engineering for the preparation or purification of various substances [24]. Under natural conditions, it is not yet sufficiently recognized. On the basis of our observations on the burning heaps of Upper Silesia, VLS can be pointed to as the main process explaining the crystallization of assemblages of anhydrous sulfate minerals in exhalations, which we will try to prove below.

Another important mineral-forming process in the discussed environment is crystallization from aqueous solutions. As these processes take place under normal pressures, the boiling points of these solutions may only slightly exceed 100 °C. Part of the water may come from the condensation of water vapor as a product of combustion and the decomposition of minerals in the fire center, but most is of meteoric origin from precipitations soaking into the porous rocky material of the heap. These solutions are acidified by sulfuric or hydrochloric acids and chemically aggressive. Their chemical composition is variable, shaped by local conditions. Various, usually hydrated, sulfate or chloride minerals crystallize from these solutions according to local hydrochemical conditions. Most of these sulfates are also known from volcanic exhalations and ore weathering zones. Some are formed by the hydration and partial dissolution of the primary, anhydrous sulfates. Their genesis can therefore be treated as a continuation of the exhalation sequence. All these sulfates are generally unstable and as the sulfate crust cools, they are gradually dissolved and carried away under the action of rainwater. What we observe in the cold sulfate crust is usually a mixture of unconverted primary sulfates and their transformation products.

The origin of the anhydrous sulfates described in this paper is variously explained by researchers. Žáček et al. [4] and Žáček and Ondruš [5] linked their crystallization to the impact of hot, chemically aggressive gases on the waste rock, and considered anhydrous sulfates to be the primary phases. These authors assumed the release of Al, Fe and other cations from waste rocks at the sulfate crystallization site close to the heap surface. Decomposition of the aluminum minerals of shales led to crystallization of aluminum sulfates. If iron minerals, e.g., siderite decomposed, iron sulfates were formed, more rarely found in this environment. A similar opinion was expressed by Stracher et al. [9] based on observations of mineralization processes in the Wuda coal-fire gas vents of Inner Mongolia, China. They claimed that "metallic cations were extracted from quartzofeldspathic sediment and rock when the gas or a liquid condensed from the gas, reacted with these materials". In their opinion, this would be a process similar to the hydrothermal leaching of metals from the substrate. We have tried unsuccessfully to find signs of chemical corrosion of fire gases

or liquids on waste rock fragments in the surface layer of the heaps we have studied. Shale pieces are red, stained by hematite, thermally baked, but no traces of chemical etching can be seen on their surface. Such etching effects should especially be seen in the sulfate crust where large amounts of anhydrous sulfates are formed. Our observations show that shale fragments are generally only a substrate on which anhydrous sulfates crystallize (Figure 8a). The extraction of components needed for their synthesis had to take place deeper and at higher temperatures. In the environment of sulfate crystallization, thermally converted fragments of shale are chemically and mechanically resistant.

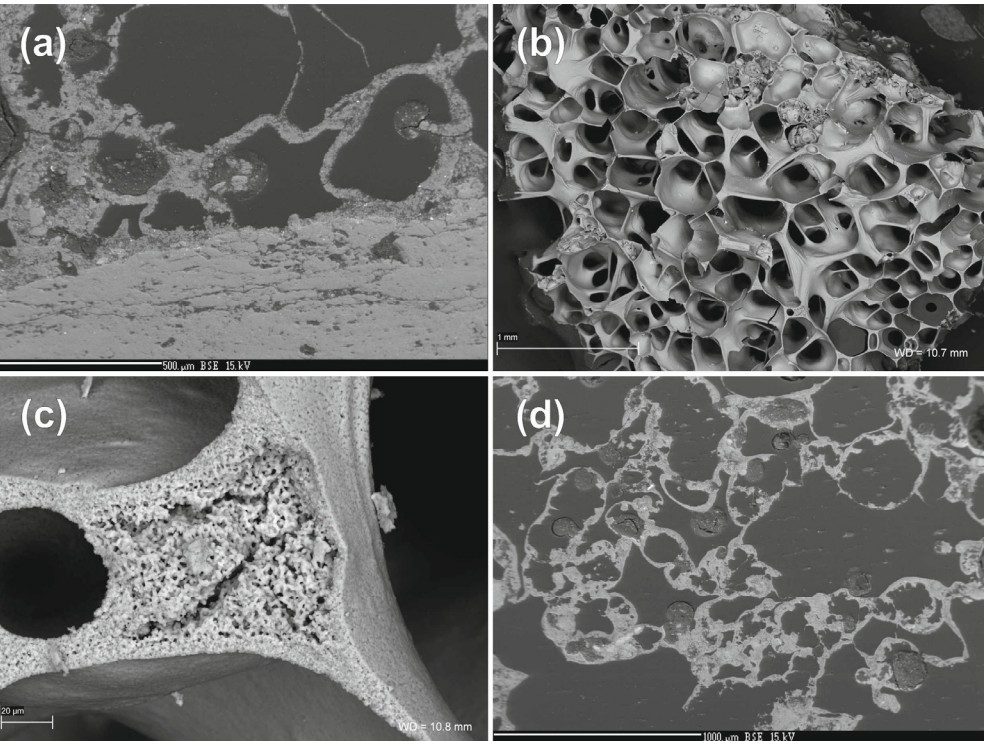

**Figure 8.** (**a**) Contact between clinker and overgrowing millosevichite foam, back scattered electron-simage (BSE), (**b**) typical vesicular structure of millosevichite (SEM), (**c**) details of microcrystalline millosevichite film around gas bubbles (SEM), (**d**) the order of microcrystalline steklite growth around gas bubbles (BSE); Rydułtowy.

Stracher et al. [9] and then Witzke et al. [21] believed that the crystallization of the discussed sulfates took place from aqueous solutions. If so, it should be assumed that these minerals were formed at a temperature slightly exceeding 100 °C, i.e., the boiling point of water solutions under normal pressure. Under such conditions, however, only hydrated sulfates such as alunogen, minerals from the alum group or boussingaultite can crystallize. According to these authors, anhydrous sulfates are therefore a product of dehydration of the primary, hydrated phases. Stracher et al. [9] associated their dehydration with an increase or fluctuation in the temperature of the fire gases. Formation of the vesicular structure characteristic of anhydrous sulfates was explained by Stracher et al. [9] by the rapid increase in temperature and the boiling of the solutions, which resulted in the escape of steam bubbles.

Based on field observations, direct measurements of the temperature of freshly formed mineral assemblage and the results of laboratory analyses, we propose a new model for the crystallization of anhydrous sulfates on burning coal-dumps of Upper Silesia. In our opinion, anhydrous sulfate minerals are of primary origin and were formed from the gas phase at temperatures much higher than previously assumed. Using an IR pyrometer, we measured the temperature of mineral accumulations in the subsurface layer of heaps at various stages of the fire inside them. The maximum values we were able to measure in

the freshly exposed sulfate crusts were 510 °C for millosevichite and steklite accumulation, and 280–360 °C for the crust formed by godovikovite. These values, although high, are lower than the real temperature of their crystallization. Our measurements were made on the already slightly cooled surfaces of mineral aggregates available to study. Therefore, they should be treated as the lower temperature range of their formation. Therefore, we do not see the possibility of the existence of water solutions under these conditions, or the crystallization of hydrated sulfates from them. How can the upper limit of the crystallization temperature of these sulfates be determined? We do not have the results of direct measurements of the temperature of fire gases inside the heap. We tried to estimate it on the basis of thermal analyses of selected, as far as possible monomineralic, samples of sulfate minerals from the heaps of Upper Silesia (Figure 9).

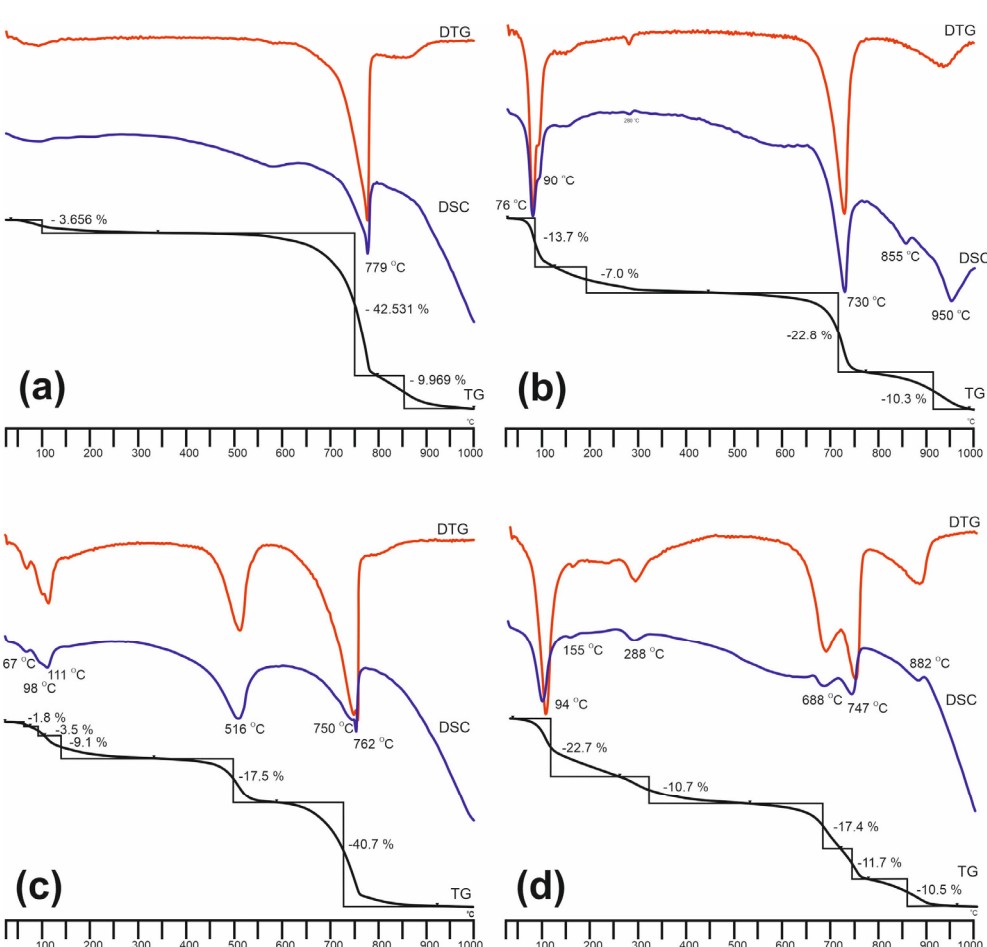

**Figure 9.** Thermal behavior of sulfate minerals. (**a**) millosevichite, (**b**) steklite, (**c**) godovikovite, (**d**) unnamed $MgSO_4$.

Only the millosevichite sample was truly anhydrous. Samples of godovikovite, steklite and unnamed $MgSO_4$ collected a few years ago were already partially hydrated during storage. However, they were anhydrous when we collected them in the field, as evidenced by the PXRD analyzes performed at that time. The degree of their hydration can be determined on the basis of low-temperature (100–200 °C) effects on DSC curves and the corresponding mass losses on TG curves. Therefore, those effects will not be taken into account in further considerations. Important for us are the temperatures of sulfate decomposition, which can be read from the thermal curves. They show that the least stable mineral of this group is godovikovite, which begins to decompose at about 450 °C. Millosevichite and steklite begin to decompose at about 650 °C and the unnamed $MgSO_4$ at about 600 °C. We can take these temperatures as the upper limit of durability of these

sulfates. Based on this, we can approximately determine the range of crystallization temperatures of godovikovite at 280–450 °C, millosevichite and steklite at 510–650 °C, and the formation of amorphous $MgSO_4$ at 510–600 °C. An even higher decomposition temperature of aluminum ammonium sulphate, 546 °C, is reported by Lopez-Beceiro et al. [25]. More or less similar crystallization temperature ranges are also given for the less frequent anhydrous sulfates in this environment which we could not directly investigate. For sabieite, Žáček et al. [4] reported a temperature limit of 115–350 °C. Mikasaite in the type locality crystallized from exhalation at about 300 °C [26]. The rhombohedral $Fe_2(SO_4)_3$ forming from the thermal decomposition of $FeOHSO_4$ appears above 400 °C and is the only phase up to 490 °C. Its breakdown starts from about 570 °C and ends at around 700 °C [27]. Values between 180–400 °C were given by Shcherbakova and Bazhenova [23] for efremovite from the type locality. Even higher crystallization temperatures are reported for aphthitalite from volcanic exhalations. Africano et al. [28] showed that it can crystallize from volcanic gases during cooling from ca. 800 °C down to 400 °C. This mineral is also known from high-temperature fluid inclusions in magmatic rocks [29,30].

The most probable mechanism for the formation of the described sulfates in the environment of burning heaps is VLS crystallization. They crystallized from hot fire gases as a result of their cooling in the near-surface layer of the heap. The components necessary for their formation were transported in the form of volatile compounds in the gas phase, whose chemical composition is unknown and requires further detailed studies. The experimental studies of the composition of gases carried out on heaps of Upper Silesia by Kruszewski et al. [31] cannot explain this because they were limited to exhalations of too low temperatures. Undoubtedly, the exhalations known from the heaps have diverse and dynamically changing chemistry, depending on the mineral composition of the rocks in the fire zone, oxygen fugacity and temperature. The composition of the gases, parental to the sulfate assemblage, was probably formed in the center of the fire at temperatures exceeding 1000 °C. Under such extreme conditions, it is possible to extract Al, Mg, Fe and alkalies from rocks and then form their volatile compounds. Important components of these gases were also ammonia and sulfur compounds resulting from oxidation of organic matter and pyrite. We suppose that the volatile metal compounds were oxycomplexes, but further research is needed.

An interesting aspect is the ability to transport aluminum in the form of aluminum sulfide at a temperature of 1000 °C [24]. The freshly exposed, hot accumulations of millosevichite, steklite, godovikovite or their mixtures from Rydułtowy often have a distinct yellow color which disappears quickly upon cooling. The cause of this fading color is unclear. It may be due to an admixture of very unstable aluminum sulfide forming at high temperatures, which, after cooling down, quickly oxidizes to sulfate. However, it disappears so quickly that its presence cannot be directly proven.

Crystallization of high temperature sulfates was not a sublimation. Sulfates were released from the gases as viscous liquid masses. This is indicated by the aggregates of godovikovite quenched in the form of gravitationally dripping masses (Figure 3). The crystallization of sulfates took place very quickly, which resulted in their cryptocrystalline shape and often the formation of very fine, impossible to separate, mixtures of these minerals. It is also indicated by the microprobe chemical analyses (Table 1). Even areas a few μm in size are not chemically homogeneous, but are subtle intergrowths of a few sulfates. This practically does not allow us to calculate the results of analyses into crystallochemical formulas of the study minerals.

Rapid crystallization also explains the formation of their typical vesicular structure, when the solidified, gas-saturated, liquid or semi-liquid mass closed the gas bubbles in the form of a thin film. The SEM image shows that first the outer zone of such a film solidifies, and then the masses located inside crystallize (Figure 8c,d).

## 5. A Separate Genetic Type of Sulfate Minerals

Assemblages of high temperature anhydrous sulfates deserve to be distinguished as a separate genetic type among sulfate minerals. They are characterized by high crystallization temperatures reaching values of 500 °C and more, unique among sulfate minerals. These minerals are typical of high temperature volcanic exhalations, but are usually found in even higher amounts in exhalations on the burning coal-dumps. Some of them, like aphthitalite and langbeinite, also crystallize in a supergene environment and are known from evaporitic deposits. All these minerals are characterized by a very dense packing structure not only among sulfates but also others oxysalts.

Accurate identification of these minerals is made difficult by the fact that they usually form very fine crystalline mixtures. Such pairs of minerals as millosevichite-mikasaite, godovikovite-sabieite, and efremovite-langbeinite being isostructural can also form solid solutions, the composition of which depends on the local geochemical and thermal conditions. High temperature sulfate minerals known from the Upper Silesian burning coal-dumps are listed in the Table 2. This list is probably not complete and further research should supplement it with further anhydrous sulfates.

**Table 2.** Assemblages of high temperature sulfates from the Upper Silesian burning coal-dumps.

| Mineral | Crystallographic System | Crystallization Temperature Range (°C) | References |
| --- | --- | --- | --- |
| Steklite $KAl(SO_4)_2$ | trigonal | 510–650 | this work |
| Millosevichite $Al_2(SO_4)_3$ | trigonal | 510–650 | this work |
| Mikasaite $Fe_2(SO_4)_3$ | trigonal | 300 | [2] |
| Godovikovite $NH_4Al(SO_4)_2$ | trigonal | 280–450 (546) | this work [25] |
| Sabieite $NH_4Fe(SO_4)_2$ | trigonal | 115–350 | [3] |
| Unnamed $MgSO_4$ | amorphous | 510–600 | this work |
| Efremovite $(NH_4)_2Mg_2(SO_4)_3$ | cubic | 180–400 | [4] |
| Langbeinite $K_2Mg_2(SO_4)_3$ | cubic | | |
| Aphthitalite $(K,Na)_3Na(SO_4)_2$ | trigonal | 400–800 | [5] |

**Author Contributions:** J.P. wrote the manuscript, with input from R.S. Both authors have read and agreed to the published version of the manuscript.

**Funding:** This work was supported by Department of Geochemistry, Mineralogy and Petrology of Faculty of Geology (University of Warsaw) (grant No. 501-D113-01-1130101).

**Conflicts of Interest:** The authors declare no conflict of interest.

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
