# Peer review of "High Temperature Sulfate Minerals Forming on the Burning Coal Dumps from Upper Silesia, Poland"

_minerals, doi:10.3390/min11020228_

Round 1

Reviewer 1 Report

See attached file.

Author Response

Overall, the manuscript is a standard mineralogical paper. Its structure, however, is unbalanced at some

places. I suggest leaving out obsolete general information on minerals (e.g., how their names are

derived, lists of other localities worldwide).

Answer: These minerals are so rare and little known that in our opinion background information about these phases may be useful to the reader.

Line 17: I do not understand: … condensation from the vapor’s viscous liquid mass … What is the

material: vapor or liquid? Rephrase/explain.

Answer:  done

Lines 37‐39: The sentence does not look to have correct word order.

Answer:  done

Line 74: spell out abbreviations first time they are used (i.e. SEM, EDS). Add the make and type of the

SEM.

Answer:  done

Lines 74‐75: keep spelling identical: X‐ray vs. X‐Ray – should be X‐ray

Answer:  done

Line 76: What does it mean DSH method? I do not know the meaning of this abbreviation.

Answer: re moved

Line 77: Should be CoKα (i.e., Greek alpha)

Answer: done

Lines 77‐79: Actually, the description of the experiment makes no sense. X‐ray tube generates all the

spectral lines plus white radiation and you can get CoKα doublet after the beam is conditioned by

monochromators and not before as your text implies. Correct!

Answer:  done

Lines 89‐92: Leave out detail info on crystals – it is trivial and generally known and brings no additional

value.

Answer:  done

Line 143: Place period after condition to end the sentence. Replace preposition “in” with “under” before word “ambient”

Answer:  done

Line 161: “More rare”? Rarer

Answer:  done

Table 1: Consider rearrangement of the columns. I believe that oxides representing anionic groups would fit better in frontmost columns. Replace comma with decimal point for the total of the last godovikovite analysis.

Answer:  Leave the table layout unchanged. The remaining corrections were made.

Line 198: better “strongly”

Answer:  done

Line 202: it hardly can dehydrate if source is anhydrous whereas product hydrous. Actually, it hydrates.

Answer: of course

Line 216: If it forms, though sporadically, monomineralic aggregates, why no chemical data for this material are provided?

Answer: Due to the rapid hydration, it was impossible to perform correct chemical analyzes of this phase.

Line 222: the fact that the phase is amorphous should be provided earlier in its description. I makes it identification dubious and definition as a mineral impossible. Check the rules under which natural substance may represent a mineral! As long as the presence of the phase is inferred solely from microprobe data I even fear whether it exists at all.

Answer: We currently know of several amorphous minerals approved by IMA(e.g. georgeite). The existence of amorphous MgSO4 was confirmed by PXRD and SEM-EDS.

Line 312: solutions containing dissolved minerals have usually boiling point above 100°C so your statement is not necessarily true.

Answer: For example : boiling point of saturated solutions of MgSO4 is 108 °C, (NH4)2SO4 – 108.2 °C, K2SO4 – 102.1 °C. The sentence was improved.

Line 326: Žaček should be written as Žáček

Answer: corrected

Line 326: “… and linked …” sounds weird, rephrase

Answer: corrected

Line 330: Rephrase the sentence beginning with “Decomposition”

Answer: corrected

Line 354: See my comment to line 312. Address this issue – boiling temperature of solutions with contents of various salts is higher than that of pure water.

Answer: done

Lines 371‐76: I do not understand. What do you mean with “real temperature” and “previously formed sulfates”?

Answer: corrected

Line 399: Žaček should be written as Žáček

Answer: corrected

Line 431: possibly “quenched” would be better than “frozen”

Answer: corrected

Table 2: Correct formulae for godovikovite and sabieite – a group NH4 should be in parentheses like in the case of efremovite

Answer: Both these forms are correct but we prefer the shorter one.

Reviewer 2 Report

Jan Parafiniuk and Rafał Siuda submitted a manuscript on the investigation of anhydrous sulphates sampled at the burning coal dumps from Upper Silesia, Poland.

First of all, I would like to explain that I looked at the manuscript with the eyes of an analytical chemist/spectroscopist who is experienced in the analysis of minerals, and I hope that other Referees will judge the work from the mineralogy side. From my point of view, I appreciate that, as far as I can see, all necessary details on the analytical measurements are given in section 2 (including details on calibration and limits of detection of wavelength-dispersive spectroscopy (to be precise, this should be wavelength-dispersive X-ray spectroscopy)), and the analyses all seem correct.

The authors analyse a group of minerals that has been studied very rarely before. Only for two exceptions (godovikovite, langbeinite) spectroscopic and diffraction data is listed in the rruff library (http://rruff.info). Only this tells me that it was worth doing this study.

The manuscript is very well written and gives comprehensive background information on every studied mineral. The authors employ a reasonable approach to estimate the lower and upper limits of their formation, and develop a hypothesis on the according mechanism, which is discussed in comparison with other potential hypotheses. This is exactly, how science works: produce robust data and develop hypotheses, which are valid until their refinement or refutation. It is a pleasure to read such manuscript.

Just to contribute to the improvement of the manuscript I mention a few typos and minor corrections:

Page 1, line 42: “… forming on the burning coal dumps …”, remove “the” as the sentence has a general meaning.

Page 3, line 94: “… Ca (diopside, TAP, Kα, 0.04), (Mg (diopside, TAP, Kα, 0.02) …”, remove the “(“ before Mg.

Page 5, line 143: “… conditions Samples …”, “.” missing.

Page 11, line 326: “… Žaček et al. [4] and Žaček and Ondruš [5] and linked …”, remove the last “and”.

Page 11, line 330: “… aluminum minerals of common there shales …”, remove “there”.

Page 13, line 397: “… MgSO4 …”, “4” should be subscript.

One comment for future work. I agree that some minerals in these mixtures seem very small for a successful Raman analysis, but Raman maps with small raster grid distances might help here. Also the fact that almost all minerals are missing in both, the ruff library and according literature, demonstrates that it would be worth trying, because this spectroscopic data would significantly contribute to the analysis of minerals by Raman spectroscopy (perhaps also in other contexts).

Author Response

First of all, I would like to explain that I looked at the manuscript with the eyes of an analytical chemist/spectroscopist who is experienced in the analysis of minerals, and I hope that other Referees will judge the work from the mineralogy side. From my point of view, I appreciate that, as far as I can see, all necessary details on the analytical measurements are given in section 2 (including details on calibration and limits of detection of wavelength-dispersive spectroscopy (to be precise, this should be wavelength-dispersive X-ray spectroscopy)), and the analyses all seem correct.

The authors analyse a group of minerals that has been studied very rarely before. Only for two exceptions (godovikovite, langbeinite) spectroscopic and diffraction data is listed in the rruff library (http://rruff.info). Only this tells me that it was worth doing this study.

The manuscript is very well written and gives comprehensive background information on every studied mineral. The authors employ a reasonable approach to estimate the lower and upper limits of their formation, and develop a hypothesis on the according mechanism, which is discussed in comparison with other potential hypotheses. This is exactly, how science works: produce robust data and develop hypotheses, which are valid until their refinement or refutation. It is a pleasure to read such manuscript.

Just to contribute to the improvement of the manuscript I mention a few typos and minor corrections:

Page 1, line 42: “… forming on the burning coal dumps …”, remove “the” as the sentence has a general meaning.

Done

Page 3, line 94: “… Ca (diopside, TAP, Kα, 0.04), (Mg (diopside, TAP, Kα, 0.02) …”, remove the “(“ before Mg.

Done

Page 5, line 143: “… conditions Samples …”, “.” missing.

Done

Page 11, line 326: “… Žaček et al. [4] and Žaček and Ondruš [5] and linked …”, remove the last “and”.

Done

Page 11, line 330: “… aluminum minerals of common there shales …”, remove “there”.

Done

Page 13, line 397: “… MgSO4 …”, “4” should be subscript.

Done

One comment for future work. I agree that some minerals in these mixtures seem very small for a successful Raman analysis, but Raman maps with small raster grid distances might help here. Also the fact that almost all minerals are missing in both, the ruff library and according literature, demonstrates that it would be worth trying, because this spectroscopic data would significantly contribute to the analysis of minerals by Raman spectroscopy (perhaps also in other contexts).

Reviewer 3 Report

This MS (# Minerals-1107820) deals with the characterization of an assemblage of anhydrous sulfate minerals formed from hot fire gases on the burning coal dumps located in the Upper Silesia, Poland.  The investigation has been carried out by means of powder X-ray diffraction, SEM-EDS, EPMA and TGA/DSC measurements.  I think this work should be published, after minor/moderate revision. Below there are some comments:

  • The paper is clear and properly organized. Nevertheless, it requires the revision of an English-speaking colleague. In particular, the sentence in lines 285-287 needs to be rewritten.
  • Some details (Lines 89-92) about standards for microprobe calibration are unnecessary, especially the one concerning PC0 one dimensional photonic crystal which has not been used. Delete lines 89-92.
  • The Authors claim to have analyzed both fresh and archival samples, but most of the presented powder diffraction data and all the TG/DT analyses (Figs. 7, 9) refer only to hydrated sulfates. Indeed, the presented TGA/DTA correspond to literature thermal analysis of tschermigite, epsomite, etc. The Authors should reconsider the temperature range of stability of anhydrous sulfates on the basis of recent published works. For instance: ammonium aluminum sulfate NH4Al(SO4)2 decomposes around 546 °C (Lopez-Beceiro et al. 2011, DOI 10.1007/s10973-010-1189-7); MgSO4 crystallizes at T>276°C (van Essen et al. 2008, https://www.researchgate.net/publication/283069842_Materials_for_thermochemical_storage_characterization_of_magnesium_sulfate);  Mikasaite Fe2(SO4)3 starts to decompose at about 570°C (Ventruti et al. 2020, Phys and Chem of Minerals, https://doi.org/10.1007/s00269-020-01113-7).
  • Delete “and” in “Žaček et al. [4] and Žaček and Ondruš [5] and linked their crystallization” (line 326).

Author Response

The paper is clear and properly organized. Nevertheless, it requires the revision of an Englishspeaking

colleague. In particular, the sentence in lines 285-287 needs to be rewritten.

Answer: done

Some details (Lines 89-92) about standards for microprobe calibration are unnecessary, especially

the one concerning PC0 one dimensional photonic crystal which has not been used. Delete lines 89-

92.

Answer: done

The Authors claim to have analyzed both fresh and archival samples, but most of the presented

powder diffraction data and all the TG/DT analyses (Figs. 7, 9) refer only to hydrated sulfates.

Indeed, the presented TGA/DTA correspond to literature thermal analysis of tschermigite,

epsomite, etc. The Authors should reconsider the temperature range of stability of anhydrous

sulfates on the basis of recent published works. For instance: ammonium aluminum sulfate

NH4Al(SO4)2 decomposes around 546 °C (Lopez-Beceiro et al. 2011, DOI 10.1007/s10973-010-1189-

7); MgSO4 crystallizes at T>276°C (van Essen et al. 2008,

https://www.researchgate.net/publication/283069842_Materials_for_thermochemical_storage_ch

aracterization_of_magnesium_sulfate); Mikasaite Fe2(SO4)3 starts to decompose at about 570°C

(Ventruti et al. 2020, Phys and Chem of Minerals, https://doi.org/10.1007/s00269-020-01113-7).

Answer: thank you for these valuable remarks. All three works are now included in the text

Delete “and” in “Žaček et al. [4] and Žaček and Ondruš [5] and linked their crystallization” (line

326).

Answer: done